# Enhancing the Adaptability of Tea Plants (*Camellia sinensis* L.) to High-Temperature Stress with Small Peptides and Biosurfactants

**DOI:** 10.3390/plants12152817

**Published:** 2023-07-29

**Authors:** Hao Chen, Yujie Song, He Li, Shah Zaman, Kai Fan, Zhaotang Ding, Yu Wang

**Affiliations:** 1Tea Research Institute, Qingdao Agricultural University, Qingdao 266000, China; chenhaotea@163.com (H.C.); songyjtea@163.com (Y.S.); lihetea@163.com (H.L.); fankaitea@163.com (K.F.); 2Tea Research Institute, Shandong Academy of Agricultural Sciences, Jinan 250100, China; shahzamantea@163.com

**Keywords:** *Camellia sinensis*, high temperature, small peptides, biosurfactants

## Abstract

Tea plants are highly susceptible to the adverse effects of a high-temperature climate, which can cause reduced yield and quality and even lead to plant death in severe cases. Therefore, reducing the damage caused by high-temperature stress and maintaining the photosynthetic capacity of tea plants is a critical technical challenge. In this study, we investigated the impact of small oligopeptides (small peptides) and surfactants on the high-temperature-stress tolerance of tea plants. Our findings demonstrated that the use of small peptides and surfactants enhances the antioxidant capacity of tea plants and protects their photosynthetic system. They also induce an increase in gibberellin (GA) content and a decrease in jasmonic acid (JA), strigolactone (SL), auxin (IAA), and cytokinin (CTK) content. At the same time, small peptides regulate the metabolic pathways of diterpenoid biosynthesis. Additionally, small peptides and surfactants induce an increase in L-Carnosine and N-Glycyl-L-Leucine content and a decrease in (5-L-Glutamyl)-L-Amino Acid content, and they also regulate the metabolic pathways of Beta-Alanine metabolism, Thiamine metabolism, and Glutathione metabolism. In summary, small peptides and surfactants enhance the ability of tea plants to resist high-temperature stress.

## 1. Introduction

With the onset of global warming, the occurrences of extreme high-temperature weather are becoming more frequent. On 13 September 2022, the World Meteorological Organization released a multi-agency report titled “United in Science”, which highlighted that the last seven years (2015–2021) were the warmest recorded and predicted a further rise in global temperatures over the next five years. Extreme high-temperature weather can have a significant impact on plant growth and development, such as reducing plant photosynthesis [1], causing male sterility [2], decreasing yield [3], increasing the likelihood of disease and pests [4], and lowering quality [5]. The tea plant, an evergreen leafy crop, typically tolerates a maximum temperature of around 34–40 °C, with a critical survival temperature of 45 °C. Unfortunately, new shoot growth is stunted when the average daily temperature exceeds 30 °C. Extreme heat waves resulting from global warming could significantly reduce tea production. It is estimated that regions in southeast China, particularly those along the Yangtze River (at approximately 30° north latitude), could experience reductions in tea yields ranging from 14% to 26% due to high-temperature stress [6]. Therefore, it is crucial to enhance the adaptability of tea plants to extreme high-temperature weather to mitigate losses.

Numerous recent studies have demonstrated the widespread involvement of small peptides as signaling molecules in the regulation of plant growth, development, immunity, and environmental adaptation [7]. Specifically, the CLAVATA3/ESR posttranslational modification of small peptides (CLE) hormone, as an extracellular signaling molecule, is widely involved in plant growth and development and responds to environmental stimuli, playing a central role in the regulation of different stem cells’ homeostasis [8]. Additionally, the small peptide hormone CLE42 acts as a negative regulator of leaf senescence. The *cle42* mutant exhibits a premature senescence phenotype, while overexpression of CLE42 (*CLE42ox*) significantly delays the senescence process. Furthermore, exogenous application of the CLE42 mature body peptide (CLE42p) can protect leaves and significantly delay leaf senescence [9]. A signaling pathway composed of small phytocytokines (SCREWs) (small peptides) and their receptor, the kinase plant screw unresponsive receptor (NUT), inhibits the process of abscisic acid (ABA) and microbe-associated-molecular-pattern (MAMP)-induced stomatal closure and can reopen stomata [10]. Small peptides can also sense the signal of water shortage in the root and transport it to the leaves through the vascular bundle for a long distance, regulating the generation of ABA and the closure of stomata in response to drought stress [11,12]. In addition to plant hormones, small peptides can mediate the balance between plant growth and stress response [13]. Some studies have also indicated the potential of small peptides in improving crop agronomic traits, suggesting that the external application of a mixture of several small peptides can enhance crop development and resistance to stress, which is convenient and has important practical applications [14]. Despite these findings, the application of small peptides for tea plant stress resistance has not yet been reported, especially regarding the mechanism of small peptides in tea plant response to high-temperature stress, which remains unclear.

Ensuring adequate wetting of the tea plant surface is critical when applying foliar fertilizer. However, the presence of a wax layer on tea leaves can make it difficult to wet them with water alone. To overcome this challenge, surfactants have become increasingly prevalent in agriculture. These compounds not only enhance the efficiency of fertilizers and pesticides, but also create a protective film on the plant surface. This layer not only helps prevent damage to cellular membranes caused by high-temperature stress, but also improves plants’ antioxidant capacity [15,16,17,18]. Despite these benefits, further research is necessary to determine whether a combination of small peptides and surfactants could more effectively enhance tea plants’ resistance to high-temperature stress.

In the search of environmentally friendly and straightforward protection measures, we combined small peptides with various surfactants, aiming to enhance tea plants’ ability to withstand high-temperature stress. We hypothesized that biosurfactants would be more effective than chemical surfactants. This study not only elucidated the mechanism of small peptides’ response to high-temperature stress in tea plants, but also provides theoretical support for the application of small peptides and surfactants for enhancing tea plants’ resistance to high temperatures.

## 2. Results

### 2.1. Effects of Small Peptides and Surfactants on Stress-Resistance-Related Indexes of Tea Leaves under High-Temperature Stress

This study investigated how small peptides and surfactants affected stress-resistance-related indicators in tea leaves under high-temperature stress and temperature recovery across various treatment groups. After subjecting the tea leaves to three days of high-temperature stress, the untreated control group showed yellowing and slight scorching in the second leaf, while the damage observed in each treatment group was relatively minor. The leaves had a glossy appearance with a slightly sticky texture. After six days of high-temperature stress, the chlorosis in the control group was more severe, with some leaves appearing irreversibly scorched. The leaves of each treatment group displayed some degree of damage, with the P0 group being less resilient. However, each treatment group had brighter leaves compared to the control group (Figure 1). Additionally, even after three days of temperature recovery, the color contrast between the control group and each treatment group remained significant, with a decrease in leaf brightness observed in each treatment group. We propose that the tea leaves partially absorbed the small peptides and surfactants. Lastly, the survival rates of each group after the test were as follows: CK 33.33 ± 5.77% b, P0 41.67 ± 2.89% b, PT 53.33 ± 5.77% a, PR 60.00 ± 5.00% a, and PS 61.67 ± 2.89% a. These results suggest that applying small peptides and surfactants can improve the phenotypic resilience of tea leaves under high-temperature stress.

In this study, we aimed to determine how small peptides and surfactants affect stress resistance in tea leaves by measuring the activity of POD, SOD, MDA, and SS content in the leaves before high-temperature stress, three days after high-temperature stress, six days after high-temperature stress, and three days after temperature recovery (Figure 2). Our results showed that, before high-temperature stress, there were no significant differences in POD activity, SOD activity, MDA content, and SS content among the experimental groups. After three days of high-temperature stress, all groups had a significant increase in POD activity, with PT, PR, and PS showing significantly higher values than CK and P0, while P0 had significantly higher activity than CK. Additionally, SOD activity increased in all groups, with P0, PT, PR, and PS exhibiting significantly higher values than CK and PS displaying significantly greater activity than P0. These changes were accompanied by higher MDA content in CK compared to P0 and PT and significantly higher content in PR and PS relative to CK. Similarly, SS content in CK was significantly greater than P0, PT, PR, and PS. After six days of high-temperature stress, POD activity started to decline, with CK having the most-notable decrease, while P0, PT, PR, and PS maintained significantly higher activities than CK. Similarly, SOD activity in PT, PR, and PS remained significantly higher than CK and P0, with P0 having significantly higher values than CK. Concerning MDA content, CK had higher values than each treatment group, with CK being significantly higher than PS. There were no noticeable differences in SS content among the treatment groups. Following three days of temperature recovery, POD activity gradually increased in all groups, with significant differences observed between the groups. Notably, P0, PT, PR, and PS had significantly greater SOD activity than CK, with no significant differences observed in MDA and SS content among the groups, respectively.

### 2.2. Effects of Small Peptides and Surfactants on Photosynthetic Indexes of Tea Leaves under High-Temperature Stress

To examine the impact of small peptides and surfactants on tea leaf photosynthesis following high-temperature stress, we measured the chlorophyll a (Chl a) content, chlorophyll b (Chl b) content, the chlorophyll a/b (Chl a/b) ratio, carotenoid (Caro) content, and the maximum photochemical efficiency of photosystem II (Fv/Fm) of the leaves before and after high-temperature stress, as well as after a period of temperature recovery (Figure 3). Prior to high-temperature stress, there were no significant differences in Chl a content, Chl b content, the Chl a/b value, Caro content, and the Fv/Fm value among the treatment groups. In fact, after three days of high-temperature stress, there was no significant difference in Chl a content between groups, while CK exhibited a higher Chl b content than P0, though this difference was not significant. CK and P0 had significantly higher Chl b content than PT and PS, with CK showing significantly higher Chl b content than PT, PR, and PS. The Chl a/b value of each treatment group was higher than CK, but the difference was only significant for PT compared to CK. Caro content in each treatment group was significantly higher than CK, and the Fv/Fm value in each treatment group was significantly higher than CK. However, after six days of high-temperature stress, the Chl a content in each treatment group was higher than CK, but only P0 showed a significant increase over CK. The Chl b content in each treatment group was lower than CK, with PR and PS exhibiting significantly lower values than CK and P0. The Chl a/b value in each treatment group was higher than CK, with PS and PR exhibiting significantly higher values than CK. The Caro content in each treatment group was significantly higher than CK, and the Fv/Fm value in each treatment group was significantly higher than CK. Following three days of temperature recovery, the Chl a content in each treatment group was significantly higher than CK, while the Chl b content in each treatment group was significantly lower than CK. The Chl a/b value in each treatment group was significantly higher than CK. Furthermore, the Caro content in each treatment group was higher than CK, but this difference was not significant. The Fv/Fm value in each treatment group was significantly higher than CK. Overall, our findings imply that the use of small peptides and surfactants can slow down the decomposition of photosynthetic pigments in tea leaves under high-temperature stress.

### 2.3. Effects of Small Peptides and Surfactants on Phytohormones Content in Tea Leaves under High-Temperature Stress

The effect of small peptides and surfactants on the phytohormone content of tea plants under high-temperature stress in different groups after three days was also studied (Appendix A). Interestingly, our results demonstrated significant differences in 13 hormone substances between P0 and CK, of which 9 were down-regulated and 4 were up-regulated. The down-regulated hormones were primarily related to IAA, CTK, and JA, whereas the up-regulated hormones were mainly related to GA. Similarly, we observed significant differences in 26 hormone substances between PT and CK, with 21 being down-regulated and 5 being up-regulated. The down-regulated hormones were mainly associated with ABA, IAA, CTK, JA, and SL, while the up-regulated hormones were primarily related to GA. We also noted significant differences in 29 hormone substances between PR and CK, with 20 being down-regulated and 9 being up-regulated. The down-regulated hormones were predominantly related to ABA, IAA, CTK, JA, and SL, while the up-regulated hormones were mainly associated with GA. Finally, we observed significant differences in 28 hormone substances between PS and CK, with 19 being down-regulated and 9 being up-regulated. The down-regulated hormones were largely related to ABA, IAA, CTK, JA, and SL, while the up-regulated hormones were primarily associated with GA.

Following our initial analysis, we conducted KEGG enrichment analysis on various hormones to gain further insight into the underlying metabolic pathways, as shown in Figure 4. Our findings revealed that the differential metabolic pathways between P0 and CK were significantly enriched in diterpenoid biosynthesis (ko00904, *p* < 0.05). No significantly enriched hormone metabolic pathways were found between the other treatment groups and the control group. We speculated that the use of active agents could affect the absorption and function of small peptides.

### 2.4. Effects of Small Peptides and Surfactants on Amino Acids in Tea Leaves under High-Temperature Stress

The determination of the amino acid analysis of tea plants in each group after three days of high temperature was also investigated in the present work (Appendix A, Figure 5). Our findings revealed significant differences in 9 amino acids between P0 and CK, with 4 increasing and 5 decreasing. The amino acids with increased content were L-Carnosine, N-Glycyl-L-Leucine, L-Trypophyl-L-Glutamic Acid, and Glycylphenylalanine, while the amino acids with decreased content were L-Alanine α-Amino Acid, N-Acetylaspartate, (5-L-Glutamyl)-L-Amino Acid, and N-Acetylaspartate (Figure 5A). Similarly, we observed significant differences in 12 amino acids between PT and CK, with 5 increasing and 7 decreasing. The amino acids with increased content were N-Glycyl-L-L-Leucine, L-Carnosine, L-Cystine, Glycylphenylalanine, and Glycine, while the amino acids with decreased content were L-Asparagine Anhydrous α-Aminoadipic Acid, γ-Aminobutyric Acid, L-Alanine, (5-L-Glutamyl)-L-Amino Acid, N α-Acetyl-L-Arginine, and Beta-Alanine (Figure 5B). Additionally, we found significant differences in 10 amino acids between PR and CK, with 7 increasing and 3 decreasing. The amino acids with increased content were L-Trypophyl-L-Glutamic Acid, N-Glycyl-L-Leucine, Glycylphenylalanine, L-Tyrosine, Glycine, and L-α-Aspartyl-L-Phenylalanine, while the amino acids with decreased content were L-Alanine, (5-L-Glutamyl)-L-Amino Acid, and Beta-Alanine (Figure 5C). Lastly, we observed significant differences in 12 amino acids between PS and CK, with 9 increasing and 3 decreasing. The amino acids with increased content were L-Carnosine, L-Trypophyl-L-Glutamic Acid, N-Glycyl-L-Leucine, L-Methionine, Glycylphenylalanine, L-Cystine, L-Ornithine, L-α-Aspartyl-L-Phenylalanine, and Glycine. The amino acids with decreased content were γ-Aminobutyric Acid, (5-L-Glutamyl)-L-Amino Acid, and α-Aminoadipic Acid (Figure 5D).

Following our initial analysis, we conducted KEGG enrichment analysis on the amino acids to gain further insight into the underlying metabolic pathways, as shown in Figure 6. Our findings revealed that the differential metabolic pathways between PT and CK were significantly enriched in Beta-Alanine metabolism (ko00410, *p* < 0.05, Figure 7A). Moreover, our observations indicated that the differential metabolic pathways between PR and CK were mainly enriched in Thiamine metabolism (ko00730, *p* < 0.05, Figure 7B). Lastly, we found that the differential metabolic pathways between PS and CK were mainly enriched in Glutathione metabolism (ko00480, *p* < 0.05, Figure 7C).

## 3. Discussion

Under high-temperature stress, the tea plant’s innate active oxygen protection enzyme system plays a critical role in protecting against oxidative damage caused by reactive oxygen species. This enzyme system comprises SOD, POD, and other components that exhibit an initial increase in activity followed by a subsequent decline under high-temperature stress. The active oxygen protection enzyme system plays a vital function in preventing damage to the cellular structure and function by oxygen free radicals, thereby safeguarding cells from oxidative damage [19,20,21]. Our study found that small peptides and surfactants can significantly enhance the resistance of tea plants to high-temperature stress. Each treatment group showed higher SOD and POD activities compared to the control group, indicating that small peptides and surfactants can protect tea plants by improving the high-oxidation-system activity. Under high-temperature stress, the excessive production of reactive oxygen species exceeds the scavenging capacity of the tea plant, leading to the accumulation of reactive oxygen species, membrane lipid peroxidation, elevated MDA content, and ultimately, increased cell plasma membrane permeability [22]. Changes in plasma membrane permeability serve as an indicator of cellular injury and the extent of the damage. Our results showed that the application of small peptides and surfactants reduced the MDA content of leaves under high-temperature stress compared to the control group. Therefore, small peptides and surfactants can enhance tea plant resilience to high-temperature stress and improve survival rates by strengthening the activities of POD and SOD and reducing MDA content. These findings suggest that small peptides and surfactants can be used as effective tools to protect tea plants against high-temperature stress and improve the overall productivity and quality of tea.

High-temperature stress can result in a reduction of chlorophyll content in plant leaves, causing detrimental effects on photosynthesis in tea plants and, in severe cases, causing irreversible damage to photosynthetic protein [23]. High temperature exacerbates the hydrolysis of chlorophyll and impedes its synthesis. Additionally, the destruction of the chloroplast membrane system can lead to a decrease in chlorophyll content. Under high-temperature conditions, the photochemical maximum efficiency of photosystem II significantly decreases, leading to photoinhibition. The actual utilization and transformation ability of photosystem II also decreases, resulting in the reduction of the activity of related enzymes, thereby affecting the photosynthetic rate [24,25,26,27]. Previous studies have indicated that the Chl a/b ratio is associated with a plant’s ability to resist stress. Chl a is blue-green, while Chl b is yellow-green. A higher ratio translates to greener plant leaves and a more-stable photosynthetic system [28,29,30,31]. This study aimed to investigate the impact of exogenous small peptides and surfactants on the photosynthetic system of tea plants. Compared to the control group, small peptides and surfactants increased the content of Chl a, Caro, the value of the Chl a/b ratio, and the Fv/Fm value in tea leaves. Our findings indicated that spraying small peptides and surfactants can mitigate the damage caused to chlorophyll under high-temperature stress and alleviate the impact of high-temperature stress on the photosynthesis of tea plants.

Plants utilize various strategies to survive when faced with high-temperature stress. One such strategy involves modifications to stress phytohormones, such as ABA and SA, as well as growth-promoting phytohormones such as IAA and GA, which can significantly enhance a plant’s ability to withstand high-temperature stress. However, the use of exogenous substances can impact the expression patterns of these plant phytohormones. Thus, it is crucial to consider the potential effects of such substances on plant physiology when attempting to mitigate high-temperature stress [32,33,34,35,36,37]. This study aimed to investigate the impact of exogenous small peptides and surfactants on hormone metabolism in tea plants. GA can enhance the ability of plants to resist abiotic stress by improving the antioxidant system and alleviating the membrane lipid peroxidation process in plants [38,39,40,41,42,43,44]. Moreover, GA can induce an increase in chlorophyll content by improving the morphology of photosynthetic chromatids [45]. In our experiment, the content of GA in the treatment groups was higher than that of the control group, indicating that small peptides and surfactants could up-regulate the expression of GA and enhance the ability of tea plants to resist high-temperature stress. JA functions to inhibit plant growth, promote leaf senescence, promote stomatal closure, inhibit Rubisco biosynthesis, and affect plant absorption of N and P and the transport of organic substances such as glucose [46,47,48,49,50,51,52,53]. In our experiment, the content of JA in the treatment groups was lower than that of the control group, indicating that small peptides and surfactants could down-regulate the expression of JA and delay the senescence of tea plants under high-temperature stress, playing a protective role on tea leaves. SL is also widely involved in plant stress [54,55,56,57,58,59]. Our results showed that small peptides and surfactants down-regulate the expression of SL and enhance the adaptability of tea plants under high-temperature stress.

Under high-temperature stress, plants often turn to differentially expressed amino acids as an essential osmotic regulator to resist stress. Evidence from numerous studies shows that regulating the amino acid metabolism pathway is a successful strategy to enhance plants’ viability in adverse conditions [60,61,62,63,64]. Notably, amino acids in plants enhance their drought resistance by affecting enzyme activity [65], and metabolomics provides evidence that amino acids play a crucial role in salt stress in plants [66]. The present study aimed to investigate the effect of exogenous small peptides and surfactants on amino acid metabolism in tea plants. Our results revealed that amino acids similarly benefit tea plants’ ability to resist high-temperature stress. Specifically, the content of L-Carnosine significantly increased in the treatment groups compared to the control group. This augmentation enhances cell antioxidant capacity, prolongs cell life, and delays aging by maintaining the body’s pH balance. Moreover, Thiamine metabolism with significant differences can help plants respond quickly to external stress by increasing mitochondrial oxidation status [67,68,69]. Glutathione metabolism and Beta-Alanine metabolism can also affect the ability of plants to resist stress by affecting the antioxidant system [70,71,72].

## 4. Materials and Methods

### 4.1. Plant Materials and Experimental Treatments

In the present study, the “Longjing 43” approved national tea variety was used. The tea plants were acclimatized with standard nutrition and an appropriate pH level in a growth chamber. Each treatment of 60 tea plants was divided into three groups, each group having 20 plants. The seedling substrate had a bulk density of around 3.5%, a porosity of around 75%, and a total organic matter content of around 60%. The ideal conditions for the tea plants were a 14/10 (day/night) photoperiod, 1000Lux of light, and 70% relative humidity. After 7 days of acclimation to 28 °C/24 °C (day/night), the following parameters were carried out: (i) The tea leaves were treated with water as the control (CK). (ii) The tea leaves were treated with 6 g L^−1^ small peptides (P0). (iii) The tea leaves were treated with 6 g L^−1^ small peptides and 0.4% of the chemical surfactant tween-20 (PT). (iv) The tea leaves were treated with 6 g L^−1^ small peptides and 0.4% biosurfactant rhamnolipid (PR). (v) The tea leaves were treated with 6 g L^−1^ small peptides and 0.4% of the biosurfactant sophorolipid (PS). During the experiment, the tea plants were sprayed every five days, and after three days, the temperature was set to 42 °C/36 °C in a high-temperature environment, followed by a return to normal temperature after six days. The small peptides were provided by Shandong Tianjiu Biotechnology Co., Ltd. (Heze, China), which were made of high-purity soybean protein via enzymatic digestion, separation, concentration, sterilization, and spray drying. Its molecular weight was less than 600 Da. The rhamnolipid used in the experiment was a substance with biological metabolic properties produced by *Pseudomonas* or *Burkholderia* [73]. Sophorolipid is a microbial secondary metabolite produced by *Candida* using sugar and vegetable oil as carbon sources through the fermentation process under certain conditions [74,75,76]. Rhamnolipid was provided by Shaanxi Ruijie Biotechnology Co., Ltd. (Xi’an, China), and sophorolipid was provided by Shandong Yousuo Chemical Technology Co., Ltd (Heze, China). Both biosurfactants are non-toxic, harmless, biodegradable, and edible. Moreover, their industrial production has reached a high degree of maturity. During the experiment, the leaves of the same part of each group of tea plants were taken for the determination of physiological and biochemical indexes and the contents of amino acids and hormones; only 2–3 leaves of each tea plant were taken for determination.

### 4.2. Determination of Antioxidant Enzyme Activity, MDA and SS Content, and Survival Rate

The superoxide dismutase (SOD) activity, peroxidase (POD) activity, malondialdehyde (MDA) content, and soluble sugar (SS) content were measured at various time points during and after high-temperature stress in this study. Specifically, we measured these indicators before the onset of high-temperature stress, three days after the onset of high-temperature stress, six days after the onset of high-temperature stress, and three days after returning to normal temperature. All measurements were performed according to the scientific research kit provided by Suzhou Grace Biotechnology Co., Ltd. (Suzhou, China) (the scientific research kit number was SOD: G0101W, POD: G0107W, MDA: G0109W, SS: G0501W). The survival rate was the ratio of surviving tea plants to the total tea plants after the end of the experiment. In this experiment, the death of the tea plants was defined as all leaves withered and no buds available for further growth. A total of 60 tea plants in each treatment were divided into three groups, and the survival rate of each group was counted, respectively. The single-factor variance test was used to analyze significant differences, and the significant differences are expressed by different letters (*p* < 0.05).

### 4.3. Determination of Photosynthetic Pigment and Fv/Fm

The chlorophyll a (Chl a) content, chlorophyll b (Chl b) content, chlorophyll a/b (Chl a/b) ratio, carotenoid (Caro) content, and maximum photochemical efficiency of the photosystem II (Fv/Fm) of the tea leaves were measured at different time points under high-temperature stress. Specifically, we measured these indicators before high-temperature stress, after three days, and after six days and three days afterward returned to the usual condition. The contents of Chl a, Chl b, and Caro were determined according to the scientific research kit provided by Suzhou Grace Biotechnology Co., Ltd. (Suzhou, China) (the scientific research kit number was G0601W). For the Fv/Fm efficiency, the second developed leaf was acclimated in the dark for 30 min, and the data were calculated by using an FP110-LM/D instrument (PSI, Drásov, Czech Republic).

### 4.4. Determination of Phytohormones and Amino Acids

The determination of phytohormones and amino acid content was also carried out in the present work under high-temperature stress for three days. After obtaining the variable importance in projection (VIP) based on the orthogonal partial least-squares-discriminant Analysis (OPLS-DA) model, metabolites with fold change ≥ 2 and fold change ≤ 0.5 were selected, and the difference was considered significant (Fold change = the average value of the experimental group/the average value of the control group. When the average value of the control group is 0, the fold change is infinite, expressed as Inf). For the regulation of the identified metabolites in different pathways, we used the KEGG compound database (http://www.kegg.jp/kegg/compound/) (accessed on 16 December 2022) and mapped them to the KEGG pathway database (http://www.kegg.jp/kegg/pathway.html) (accessed on 16 December 2022). We mapped the significant metabolic pathways to the metabolite concentration analysis (MSEA) dataset and assessed the significance of the results by performing hypergeometric tests using the following formula: (1)P=1−∑i=0m−1MiN−mn−iNn
where *N* represents the total number of metabolites with KEGG annotations, *n* represents the number of differential metabolites in *N*, *M* represents the number of metabolites in a specific KEGG pathway, and m represents the number of differential metabolites in *M* in the KEGG pathway. The phytohormone and amino acid content was evaluated by Maiwei Biotechnology Co., Ltd. (Wuhan, China) using the AB Scienx QTRAP 6500 LC-MS/MS platform (SCIEX, Framingham, MA, USA).

### 4.5. Statistical Analysis

Statistical analysis was conducted using the SPSS 20.0 software (SPSS Inc., Chicago, IL, USA) and the GraphPad Prism 9.4.0 software (GraphPad Software, San Diego, CA, USA). All data are expressed as the means ± the standard deviations (*n* ≥ 3). The significant differences were determined by using Duncan’s multiple-range tests (*p* < 0.05).

## 5. Conclusions

Based on our findings, we concluded that the exogenous small peptides and surfactants significantly enhanced the thermotolerance of tea plants and increased their survival rate under high-temperature stress. The effectiveness of small peptides can be further boosted by surfactants, with biosurfactants such as rhamnolipid and sophorolipid exhibiting superior performance to chemical surfactants such as tween-20. Notably, there were no significant differences observed in the performance between the two biosurfactants. The primary effects of small peptides and surfactants on tea plants were as follows: (1) enhancing the antioxidant system of tea plants by increasing the activities of POD and SOD, while reducing MDA content; (2) elevating the levels of Chl a, Caro, the Chl a/b ratio, and the Fv/Fm value, thereby protecting the photosynthetic system of the tea plants; (3) up-regulating GA expression while down-regulating the expression of JA, SL, IAA, and CTK to enhance the thermotolerance of the tea plants; (4) enhancing the thermotolerance of the tea plants by inducing an increase in amino acid content such as L-Carnosine and N-Glycyl-L-Leucine and regulating metabolic pathways such as Beta-Alanine metabolism, Thiamine metabolism, and Glutathione metabolism. Overall, these findings may have practical implications for the cultivation of tea plants under high-temperature stress.

## Figures and Tables

**Figure 1 plants-12-02817-f001:**
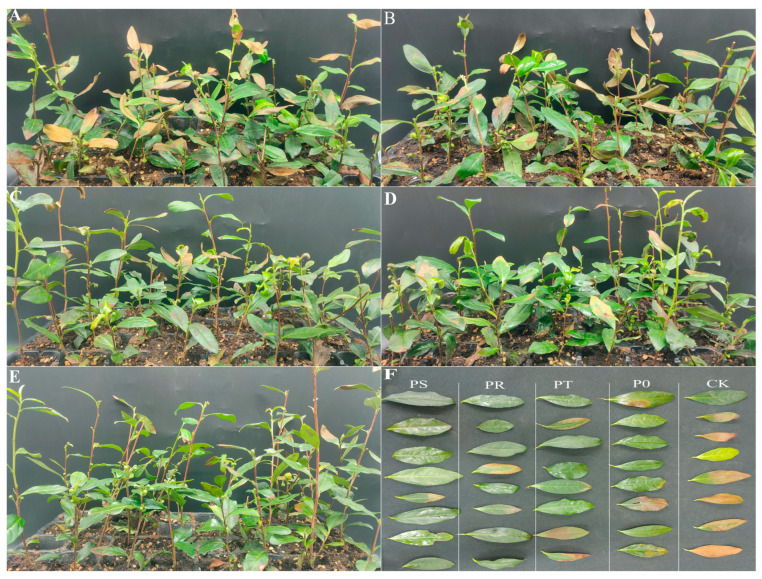
(**A**) Tea plant status treated with CK (spraying clean water on the leaf surface) after 6 days of high-temperature stress. (**B**) Tea plant status treated with P0 (spraying small peptides on the leaves) after 6 days of high-temperature stress. (**C**) The tea plant state of PT (spraying small peptides and tween-20 on the leaves) treatment after 6 days of high-temperature stress. (**D**) Tea plant status treated with PR (spraying small peptides and rhamnolipid on the leaves) after 6 days of high-temperature stress. (**E**) Tea plant status under PS (spraying small peptides and sophorolipid on the leaves) treatment after 6 days of high-temperature stress. (**F**) Leaf status of the second leaf of tea plants in each treatment group after 6 days of high-temperature stress.

**Figure 2 plants-12-02817-f002:**
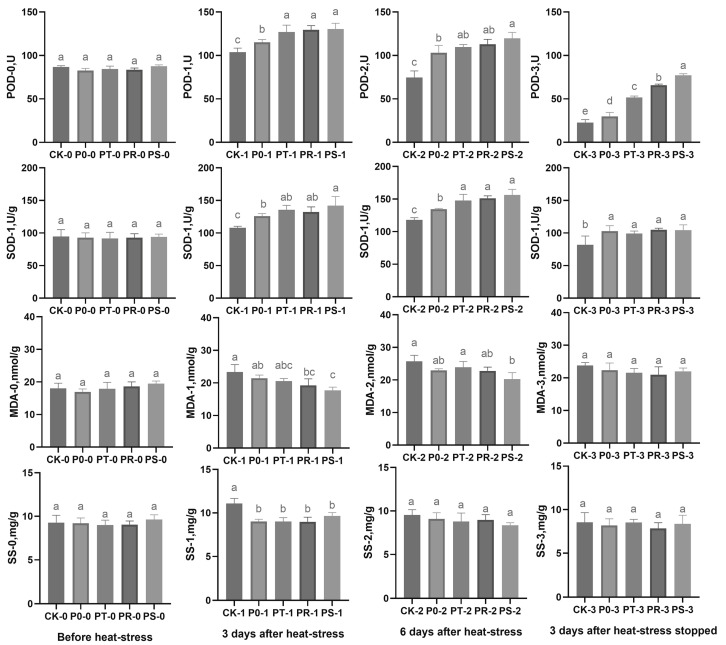
Peroxidase (POD) activity, superoxide dismutase (SOD) activity, malondialdehyde (MDA) content, and soluble sugar (SS) content of each treatment (CK, P0, PT, PR, and PS) before high-temperature stress, 3 days after high-temperature stress, 6 days after high-temperature stress, and 3 days after temperature recovery. Different letters indicate the significant difference between the control plant group and each treatment group under the given high-temperature stress time (*p* < 0.05). “0” represents before high-temperature stress; “1” represents 3 days after high-temperature stress; “2” represents 6 days after high-temperature stress; “3” represents 3 days after temperature recovery.

**Figure 3 plants-12-02817-f003:**
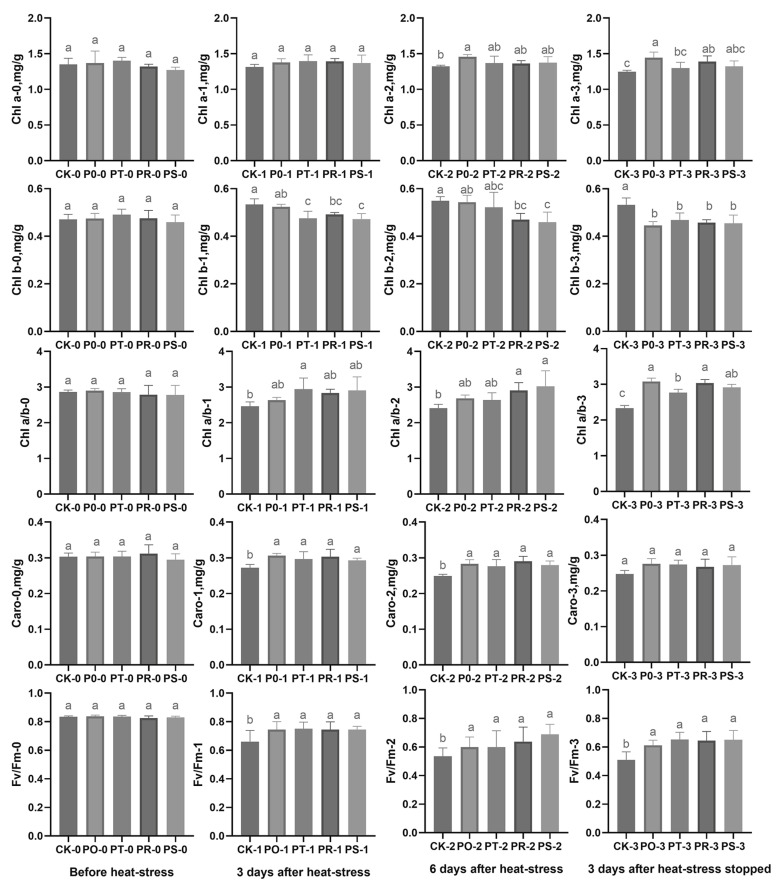
Chlorophyll a (Chl a) content, chlorophyll b (Chl b) content, chlorophyll a/chlorophyll b (Chl a/b) ratio, carotenoid (Caro) content, and Fv/Fm value among treatments (CK, P0, PT, PR, and PS) before high-temperature stress, 3 days after high-temperature stress, 6 days after high-temperature stress, and 3 days after temperature recovery. Different letters indicate the significant difference between the control plant group and each treatment group under the given high-temperature stress time (*p* < 0.05). “0” represents before high-temperature stress; “1” represents 3 days after high-temperature stress; “2” represents 6 days after high-temperature stress; “3” represents 3 days after temperature recovery.

**Figure 4 plants-12-02817-f004:**
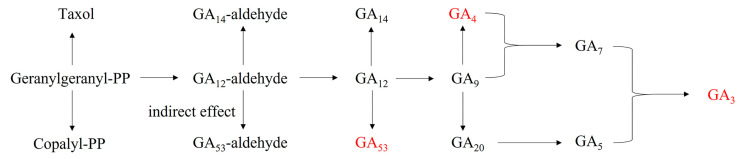
Diterpenoid biosynthesis; red represents a compound with a significant increase in content.

**Figure 5 plants-12-02817-f005:**
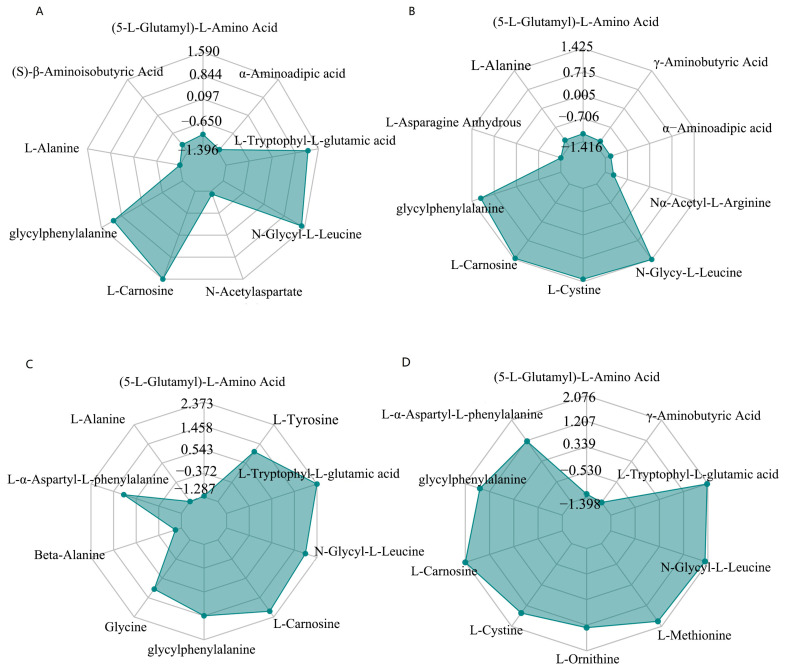
(**A**) The significant difference of amino acid expression between P0 and CK. (**B**) Differential amino acid expression between PT and CK. (**C**) Differential amino acid expression between PR and CK. (**D**) The radar chart of differential amino acid expression between PS and CK only shows the top 10 amino acids with the largest difference. The grid line corresponds to the value of the differential multiple, and the green shadow is composed of the line of the differential multiple corresponding to each substance.

**Figure 6 plants-12-02817-f006:**
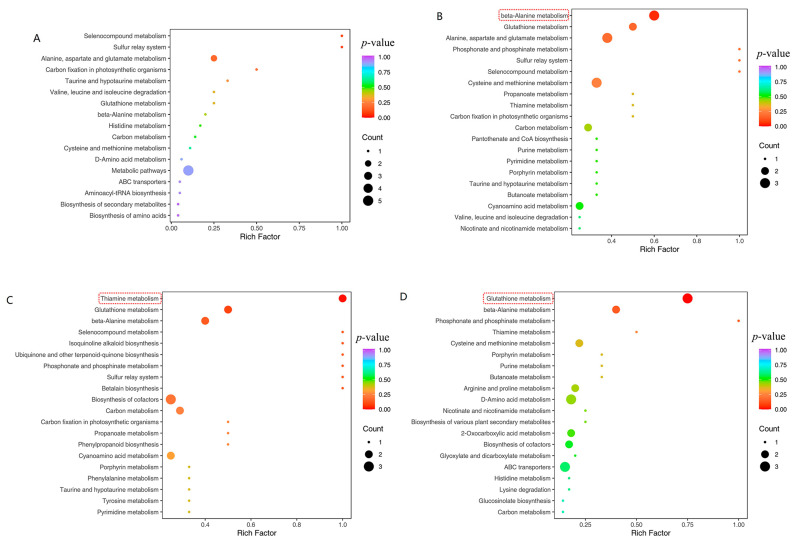
(**A**) The difference of amino acid metabolism pathway between P0 and CK. (**B**) The difference of amino acid metabolism pathway between PT and CK. (**C**) Difference of amino acid metabolism pathway between PR and CK. (**D**) The difference of amino acid metabolism pathway between PS and CK. The abscissa represents the corresponding Rich factor for each pathway. The Rich factor is the ratio of the number of differential metabolites in the corresponding pathway to the total number of metabolites annotated by the pathway. The higher the value, the greater the enrichment degree is. The ordinate is the path name, and the color of the point is the *p*-value. The redder the point, the more significant the enrichment is. The size of the dot represents the number of enriched differential metabolites. The red dotted line represents the metabolic pathway with *p* < 0.05.

**Figure 7 plants-12-02817-f007:**
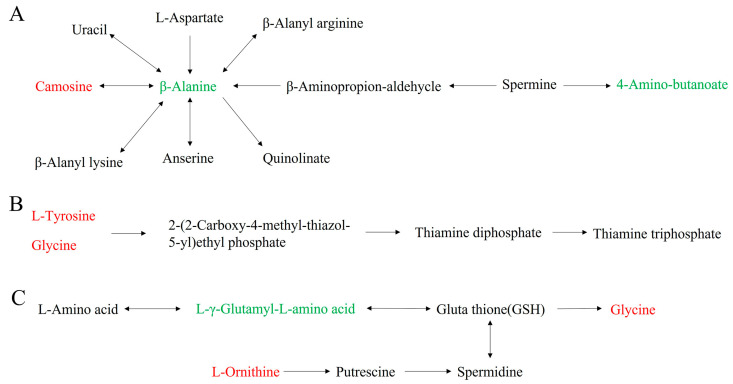
(**A**) Beta-Alanine metabolism, (**B**) Thiamine metabolism, and (**C**) Glutathione metabolism, Red represents a compound with a significant increase in content. Green represents a compound with a significantly reduced content.

## Data Availability

The data presented in this study are available within the article.

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
