# Peer review of "Enhancing the Adaptability of Tea Plants (Camellia sinensis L.) to High-Temperature Stress with Small Peptides and Biosurfactants"

_plants, 2023, doi:10.3390/plants12152817_

Round 1
Reviewer 1 Report
Review comments
Heat stress substantially reduces the productivity of crop plants including tea plants. It is a good idea to spray small peptides and surfactants for enhancing heat tolerance of tea plants. The results are publishable. The English writing is readable although it is far away from well written.
I have a few following suggestions:
(1) F values of statistical analyses should be provided.
(2) Information on the replicates, e.g., number of plants in each group and number of leaves on each plants measured, etc., should be provided.
(3) English writing issue. Line 19, “while small peptides regulating metabolic pathways of diterpenoid biosynthesis”
(4) English writing issue. Lines 21 to 22, “while regulating the metabolic pathways of beta-Alanine metabolism, thiamine metabolism and glutathione metabolism”
(5) English writing issue. Lines 73 to 74, “while also improving plants' antioxidant capacity”
(6) Lines 172 to 173 “… using small pep-172 tides and surfactants can improve the photosynthetic resistance…”. The authors did not measure CO2 fixation rates. This statement is not appropriate.
(7) Figure 4 should be improved. It is not self-explained.
(8) English writing issue. Line 285 “while reducing MDA content”
(9) Line 291 “light”. Typo?
(10) Lines 347 to 358, “artificial chamber”. Growth chamber
(11) English writing issue. Line 352, “with water signify as control”
(12) English writing issue. Line 368. “their industrial production has been very mature.” Chinse English.
(13) Line 380, “60”. A total of 60.
(14) Line 418, “this”. Typo.
(15) Line 427, “safeguarding”. Not appropriate.
See my comments above
Author Response
Dear Prof.,
Thank you for spending time reviewing our manuscript “Enhancing the Adaptability of Tea Plants (Camellia sinensis L.) to High Temperature Stress with Small Peptides and Biosurfactants (plants-2511619)” and providing us with a list of constructive comments and suggestions.
You read very carefully, thank you very much for your comments and suggestions, and made a great contribution to the quality improvement of our manuscript.
Thank you again for your valuable time.
Best regards,
Zhaotang Ding
Responses to the comments of Reviewer
Question 1: F values of statistical analyses should be provided
Answer: Thank you for your guidance. We have provided the F value of statistical analysis in the manuscript, including the formula of the F value, and each original data value required to analyze the F-value. (Line 410-412 and Table S1)
Question 2: Information on the replicates, e.g., number of plants in each group and number of leaves on each plants measured, etc., should be provided.
Answer: Thank you for your guidance. We have provided more detailed information about repeated trials, including the number of tea trees in each group, the number of leaves measured on each tea tree, etc. (Line 352-353 and Line 375-378)
Question 3: English writing issue. Line 19, “while small peptides regulating metabolic pathways of diterpenoid biosynthesis”.
Answer: Thank you for your guidance. We have corrected it. (Line 19)
Question 4: English writing issue. Lines 21 to 22, “while regulating the metabolic pathways of beta-Alanine metabolism, thiamine metabolism and glutathione metabolism”.
Answer: Thank you for your guidance. We have corrected it. (Line 21-23)
Question 5: English writing issue. Lines 73 to 74, “while also improving plants' antioxidant capacity”.
Answer: Thank you for your guidance. We have corrected it. (Line 73-75)
Question 6: Lines 172 to 173 “… using small pep-172 tides and surfactants can improve the photosynthetic resistance…”. The authors did not measure CO2 fixation rates. This statement is not appropriate.
Answer: Thank you for your guidance. We have corrected it. (Line 176-178)
Question 7: Figure 4 should be improved. It is not self-explained.
Answer: Thank you for your guidance. We have improved it. (Line 212)
Question 8: English writing issue. Line 285 “while reducing MDA content”
Answer: Thank you for your guidance. We have corrected it. (Line 289)
Question 9: Line 291 “light”. Typo?
Answer: Thank you for your guidance. We have corrected it. (Line 294)
Question 10: Lines 347 to 358, “artificial chamber”. Growth chamber
Answer: Thank you for your guidance. We have corrected it. (Line 352)
Question 11: English writing issue. Line 352, “with water signify as control”
Answer: Thank you for your guidance. We have corrected it. (Line 357)
Question 12: English writing issue. Line 368. “their industrial production has been very mature.” Chinse English.
Answer: Thank you for your guidance. We have corrected it. (Line 372-374)
Question 13: Line 380, “60”. A total of 60.
Answer: Thank you for your guidance. We have corrected it. (Line 390)
Question 14: Line 418, “this”. Typo.
Answer: Thank you for your guidance. We have corrected it. (Line 430)
Question 15: Line 427, “safeguarding”. Not appropriate.
Answer: Thank you for your guidance. We have corrected it. (Line 439)
Reviewer 2 Report
40-50 years ago similar problem was discussed as using of antitranspirants in Forestry and Agriculture, therefore heat abiotic stress is not a new problem. The represented paper is related to the great problem for tea agroculture, and may be published in Plants.
To my mind, this problem and technical rules like inappropriate self-citation or plagiarism are incompatible. We have a lot of experiments and a lot of working scientists ! I may recommend authors to estimate practical ways of the small proteins and biosurfactants application in the field conditions. I saw leaves treated by antitranspirants , and new results with the new components on leaves look better, than earlier data. This method is suitable, but there are questions for the food using : a decomposition of the reagents and danger for human health.
The and a before word in some cases, but it is not principial.
Author Response
Dear Prof.,
I would like to express my sincere gratitude for your time, effort, and expertise in reviewing my manuscript titled “Enhancing the Adaptability of Tea Plants (Camellia sinensis L.) to High Temperature Stress with Small Peptides and Biosurfactants (plants-2511619)”. Your valuable comments and constructive criticism have greatly improved the quality of my work.
Your insightful feedback and attention to detail have been truly invaluable to me. Your suggestions and recommendations have helped me to refine and clarify my ideas, and have led to a much stronger and more coherent manuscript. Based on your suggestions, we will evaluate practical methods for the application of small peptides and biosurfactants under field conditions. We anticipate that this low-cost approach may yield promising results. Additionally, you raised concerns regarding the decomposition of reagents and their potential impact on human health. The two biosurfactants mentioned in our manuscript have already been used in food products and comply fully with national safety standards.
Thank you again for your generous contribution to this work.
Sincerely,
Zhaotang Ding